

# Influence of drop size distribution and kinetic energy in precipitation modelling for laboratory rainfall simulators

Harris Ramli[1], Siti Aimi Nadia Mohd Yusoff[1], Mastura Azmi[1], Nuridah Sabtu[1], Muhd Azril Hezmi[2]

[1]School Of Civil Engineering, Engineering Campus, Universiti Sains Malaysia, 14300 Nibong Tebal, Seberang Perai Selatan, Pulau Pinang, Malaysia
[2]School of Civil Engineering, Universiti Teknologi Malaysia, 81310 Johor Bahru, Johor, Malaysia.

*Correspondence to*: Mastura Azmi (cemastura@usm.my)

**Abstract.** It is difficult to define the hydrologic and hydraulic characteristics of rain for research purposes, especially when trying to replicate natural rainfall using artificial rain on a small laboratory scale model. The aim of this paper was to use a drip-type rainfall simulator to design, build, calibrate, and run a simulated rainfall. Rainfall intensities of 40, 60 and 80 mm/h were used to represent heavy rainfall events of 1-hour duration. Flour pellet methods were used to obtain the drop size distribution of the simulated rainfall. The results show that the average drop size for all investigated rainfall intensities ranges from 3.0-3.4 mm. The median value of the drop size distribution or known as $D_{50}$ of simulated rainfall for 40, 60 and 80 mm/h are 3.4, 3.6, and 3.7 mm, respectively. Due to the comparatively low drop height (1.5 m), the terminal velocities monitored were between 63-75% (8.45-8.65 m/s), which is lower than the value for natural rainfall with more than 90% for terminal velocities. This condition also reduces rainfall kinetic energy of 25.88- 28.51 J/m²mm compared to natural rainfall. This phenomenon is relatively common in portable rainfall simulators, representing the best exchange between all relevant rainfall parameters obtained with the given simulator set-up. Since the rainfall can be controlled, the erratic and unpredictable changeability of natural rainfall is eliminated. Emanating from the findings, drip-types rainfall simulator produces rainfall characteristics almost similar to natural rainfall-like characteristic is the main target.

## 1 Introduction

Rainfall simulators are usually used to produce the artificial rainfall under controlled conditions, which is found to be useful in surface hydrology and studies of soil erosion whether in laboratory or in field. The information obtained includes infiltration characteristics, sediment yield/erosion parameters, and nutrient export indices (Ward and Bolton, 1991). Infiltration is useful to model the behaviour of soil, in particular with the soil reaction under the influence of rainfall intensities. The impact of the rainfall on soil surface causing dislodging soil particles and splashing. Most of the splashed soil, resulting in surface pores clogging which in turn reduces water infiltration, increases water runoff, and increases soil erosion. Since the rainfall play a big role in slope erosion, the development of the rainfall simulator to replicates the natural rainfall is very important aspect to look at. Although rainfall simulators are convenient in design, it creates difficulties when comparing results is required between different simulators. Rainfall simulators can provide more consistent control over an experiment, as well as faster data collection and repeatable testing. The simulators allow control of the intensity of the applied rain, are efficient in terms of time

and labour required and can be easily adapted for laboratory studies (Moussouni et al., 2014; Nampula et al., 2016). Meanwhile according to Iserloh et. al (2012), its application allows a quick, specific, and reproducible assessment of the meaning and impact of several factors, such as slope, soil type (infiltration and permeability), soil moisture, splash effect of raindrops
(aggregate stability), surface structure, vegetation cover and vegetation structure.

Rainfall simulators must attain several major characteristics of natural rainfall such as random distribution of raindrop size, speed of impact similar to the terminal velocity of natural raindrops, rainfall intensity corresponding to natural conditions, Kinetic energy similar to that of natural rain and lastly uniform rain and random distribution of drops (Mech, S.J, 1965; Hall,
1970; de Lima et al., 2013).

The influence of the kinetic energy of water droplets can be determined by varying the height of fall and sizes of the droplets or through the usage of residue cover to protect the soil surface from adsorbing droplet energy. Droplets from the simulator induced by gravity are initiated from repose and accelerate to a certain terminal velocity depending on the drop diameter.
Smaller diameter droplets have a lower terminal velocity because of their size, therefore requiring lower fall heights to achieve this terminal velocity (Regmi & Thompson, 2000). Droplet size influences the kinetic energy of droplets depending on the square of droplet velocity at impact. The kinetic energy effect is critical to the chemical transfer process and soil movement. Surely, simulated rainfall should made to be as close to natural rainfall as possible, especially on droplets' fall velocity and size distribution.


Rainfall simulators are categorised as tubing-tip (i.e., hypodermic needle or drop forming type simulators) and nozzle type simulators (Regmi & Thompson, 2000). A nozzle-type simulator increases the working pressure and consequently the average intensity of the rainfall produced. However, the range and drop size mean decreases within the spray. The drawback of this nozzle type is that it produces raindrops contrary to natural rainfall. As an alternative, tubing-tip simulators using fixed tube
diameters produce uniform drop size droplets. There are two types of rain simulators based on the mechanism of droplets, selected according to their availability: cost of construction and the experimental objective. The drip method, where the initial velocity of the droplets is zero as it is not a pressurized system (Chen et. al, 2018) and has a relatively low cost. However, the desired final velocity is achieved at drop heights of 12 m and in larger diameter droplets (Kathiravelu et al., 2016).

The spray nozzle mechanism is when the water exits at an initial velocity different from zero because it is subjected to a determined initial pressure (Nampula et al., 2016). This simulator can provide rainfall of different intensity so that it is possible to simulate the characteristics of natural rainfall according to the study area (Navas et al., 1990; Cerdà et al., 1997; Abudi et al., 2012; Nampula et al., 2016). The problem with this simulator is that very high and unnatural intensities are required to obtain droplet sizes similar to those of natural rainfall, so they require mechanisms that allow them to be diminished while
preserving the dimensions of the droplets. Rotary discs with a radial groove, a nozzle in an oscillating system, and oscillating



motion sprinklers have been used in Mexico (Paige et al., 2004; Nampula et al., 2016). Another important feature is the size of the raindrop, as this will influence the intensity of the raindrop and the final kinetic energy. In designing rainfall simulators, several considerations must be considered, such as the physical and mechanical properties of the drop former. The properties include appropriate density under constant head conditions that will influence the intensity of the raindrops based on the

spacing of drop formers, uniformity of the application and the droplet size distribution. Also, the control rate of application and terminal velocity of the droplet distribution must be considered. Therefore, this study aims to present the design and construction of a laboratory-scale rainfall simulator, capable of producing droplet characteristics identical to natural rainfall.

## 2 Design and Construction of the Rain Simulator

A laboratory-scaled rainfall simulator was constructed at Geotechnical Laboratory, Universiti Sains Malaysia. The structural

frame of the rainfall simulator is made of Dexion steel, which provides the stability for a drip system of 1.5 m in height. A 50 L water tank connected to the water supply systems supplies water to the model set-up. The tank is equipped with a pump through the main water pipe and is divided into ten lateral drip pipes to produce raindrops. Flowmeter was installed on the rainfall simulator system in order to monitor the discharge in the system. It works by measuring the amount of a liquid flowing through it. The discharge can be manually controlled using a pressure regulating valve, allowing for the desired rainfall

intensity to be achieved. The drippers used for the tests were Rainjet drippers model 91217 and 91209 manufactured by Claber. The drippers were chosen since they can produce uniformly distributed drops and sizes. They were attached to the Dexion steels of the laterals. Figure 1 (a) shows the experimental set up of a rainfall simulator meanwhile Figure 1 (b) shows the drip emitter systems consists of ten laterals connected to the sub-main. Every 1 m long lateral contained ten drip emitters equally spaced at 0.8 m. The pipe diameter is 2 cm while the spray diameter is 4 mm. The water delivery system was designed to

provide adequate flow to the risers whilst maintaining a suitable operating pressure. The constant rainfall intensities that have been chosen are 40, 60 and 80 mm/h, considered by the Department of Drainage and Irrigation, Malaysia, as heavy rainfall. Intensity is controlled using a flowmeter to supply the corresponding discharge per surface unit.



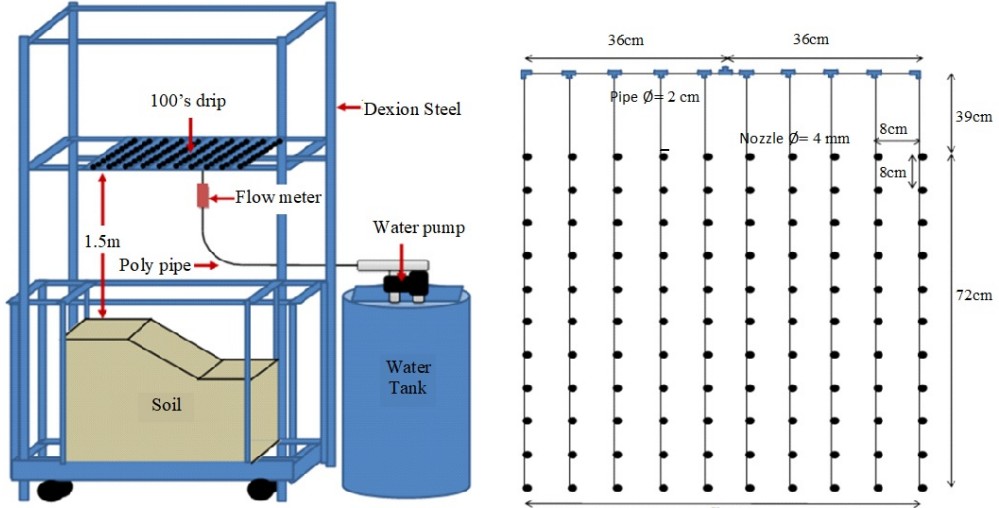

**Figure 1: (a) Components of rainfall simulator  and (b) schematic view of a drip irrigation system**

## 3 Rainfall Characteristics

### 3.1 Velocity of drops

As the raindrop falls, it accelerates due to the gravitational force simultaneously as opposed to the drag provided by the atmosphere. The raindrops travel sufficient distance to attain a constant velocity, termed as terminal velocity. Rain-drop

terminal velocity, $V_T$ is the balance between two opposites − gravitational and drag − forces acting on the drop during its vertical motion. Gunn & Kinzer (1949) have suggested an empirical expression for the terminal velocity of a water droplet (Mukhopadhyay & Tripathi, 2015). They observed that terminal velocity $V_T$ (cm/sec) of a water droplet of radius $r$ (cm) could be expressed as

$$V_T = 200\sqrt{r} \qquad\qquad\qquad (2)$$

In the experiment, it is noted that the terminal velocity depends on its diameter as bigger raindrops experience the drag to a greater extent. Foote & Du Toit, (1969) established an expression that estimates terminal velocity ($V_T$ (m/sec)) of the falling raindrop in terms of a polynomial of the raindrop diameter ($D$ (mm)). For third-order polynomial approximation, the terminal velocity is given as

$$V_T = -019274 + 4.9625D - 0.90441D^2 + 0.056584D^3 \qquad\qquad (3)$$


### 3.2 Drop Sizes

The mean size of raindrops and their percentage distribution can be used to classify the type of rainfall (Yakubu et. al, 2016). As a result, simulated precipitation must be comparable to natural precipitation to produce similar characteristics. Different





methods for estimating drop size were developed based on studies that estimate drop size from simulated or natural rainfalls. These methods include using disdrometer, photographic techniques, coloured methylene blue filter paper, and flour pellet method variants. However, in this study the flour pellet method was choose because according by Kathiravelu et al. (2016), results obtained using flour pellet method are more adequate for raindrops between sizes 0.3 and 6.0 mm in diameter. It is also important to note that high rainfall intensities degrade the quality of measurements using the disdrometers or imaging techniques, due to background noise (Mendes et al., 2021).


This study determined the drop size using the flour pellet method as proposed by Bentley (1904) (Kathiravelu et al., 2016; Chowdhury et al., 2017). This consists of a 1.0 m * 1.0 m plate containing a 2.54 cm (1 inch) layer of uncompacted fine wheat flour. The plate is exposed to rainfall for 1 to 4 seconds through manual control of the valve, subjected to the intensity of rainfall. The drops must not fall at the same point during rainfall simulation and formed corresponding to the drop size on impact. The plate containing wheat flour was covered to protect it from rainfall except when the cover was removed during the experiment to collect the drop samples. Each drop that falls on the pallet is made into small flour balls, dried for 24 hours at 105 degrees Celsius, and then sieved. The samples were sieved according to BS 1377-2:1990 Methods of Test for Soils for Civil Engineering Purposes – Classification Test, using sieve sizes of 6.30, 5.00, 4.47, 2.36, 2.00, 1.18 and 0.60 mm. The granulometric fractions obtained are weighed. The relationship between the diameter of the drop, $D_d$ (in mm), and the mass of the flour ball, $M_f$ (in mg), is $D_d = 14.56 M_f^{0.354}$, respectively. To estimate the drop diameter as a function of the flour ball diameter $D_d$ (mm) $= 0.985 D_f$ (mm)$^{1.02}$, the diameter of flour balls can be considered practically equivalent to that of the drops.

### 3.3 Kinetic energy

The kinetic energy of a raindrop is a function of the size of the raindrop and its terminal velocity. The kinetic energy ($E$) in ergs of a falling raindrop is determined as:

$$E = \frac{1}{2} mV^2 \tag{4}$$

Where $m$ is the mass of falling raindrop (g), and $v$ is the velocity of fall (cm/s). The total kinetic energy for the storm is estimated by the summation of $E$ values from individual raindrops. Measurements of $E$ of raindrops is difficult under natural rain. Therefore, Wischmeier & Smith (1958) and Van Dijk et al., (2002) proposed estimating the kinetic energy of natural rainfall using Equation 5.

$$KE = 11.897 + 8.73 \log_{10} I \tag{5}$$

Where $KE$ = Kinetic energy (J/m$^2$ mm) and $I$ = Rainfall Intensity, measurement of physical characteristics of rainfall simulated such as raindrop size, fall velocity and drop size distribution contributed to the kinetic energy. Therefore, rainfall simulators that can produce similar properties as natural rainfall would similarly produce the same kinetic energy as natural rainfall.



## 4 Results and Discussion

Simulated real intensity ($Ir$) can be linked to operating pressure that fit the potential type of function, y = 0.27x + 16.5, with the $R^2$ value of 0.9694. A direct correlation between the operating pressure and simulated rainfall intensity of the laboratory observation and the manufacturer between can be seen in Figure 2. Results indicated discharge flow rates of all 100 drips consistent with the design flow rate as postulated by the manufacturer. The relationship between discharge rate to intensity and the number of drops to intensity are shown in Table 1 and Figure 3.

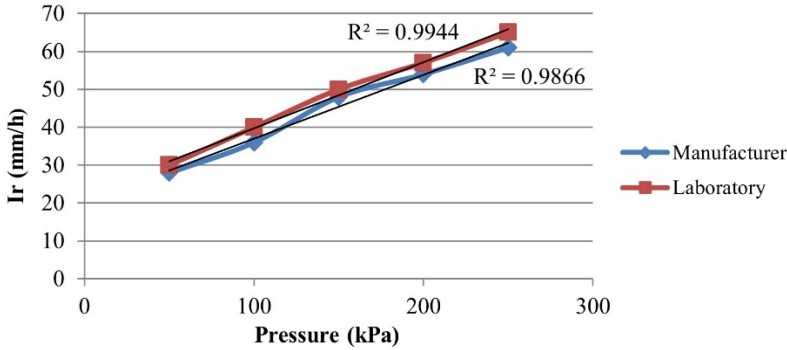

**Figure 2: Relationship between simulated rainfall and operating pressure**

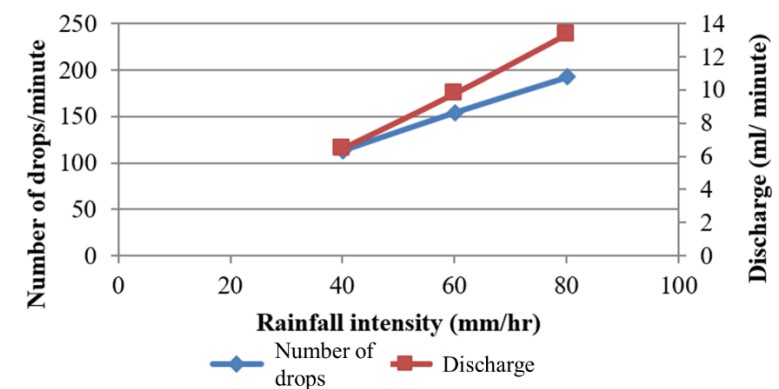

**Figure 3: Relationship of the rate of discharge and number of the drop to rainfall intensity**

**Table 1: Relationship of rate of drop formation to intensity**

| Rainfall intensity | Number of drop/minutes | Discharge (ml/ minute) |
|---|---|---|
| 40 | 113 | 6.44 |
| 60 | 154 | 9.74 |
| 80 | 192 | 13.3 |





Figure 3 shows that at 40 mm/hr, the number of raindrops per minute is 113, with 6.44 grams per minute of a discharge. Meanwhile, for 60 mm/hr, the raindrop per minute is 154 with 9.78 grams per minute of a discharge. Lastly, for 60 mm/hr, raindrops per minute are 192, with 13.3 grams per minute of a discharge. These results shows that the discharge rate and the

number of drops per minute increased as the rainfall intensity increased.

- **Drop size distribution**

A total of 100 drop samples were collected from a container filled with flour for sieving to determine the size distribution of the drops. The drop sizes are split into five classes ranged from 2.0 mm to 5.0 mm. Result through sieving shoes that the

droplet samples range between 2.0 to 4.5 mm diameter mirror the natural raindrop size. The drop size distribution is shown in Figure 4. The maximum number of drops can be observed between the range of 3.0 to 3.4 mm diameter. This size range is apparent for most heavy rainfall cases. From the rainfall classification by Varikoden et al. (2010) for Peninsular Malaysia and a case study by Yakubu et al. (2016), the results obtained in this study scan be categorised as heavy rainfalls that produce large volumes of rain within a short duration. The raindrop size distribution curves are presented in Figure 5, where the raindrops

median diameter is obtained and presented. The median diameter value ranges from 3.0 to 3.3 mm for the 40, 60, 80 mm/hr rainfall intensities.

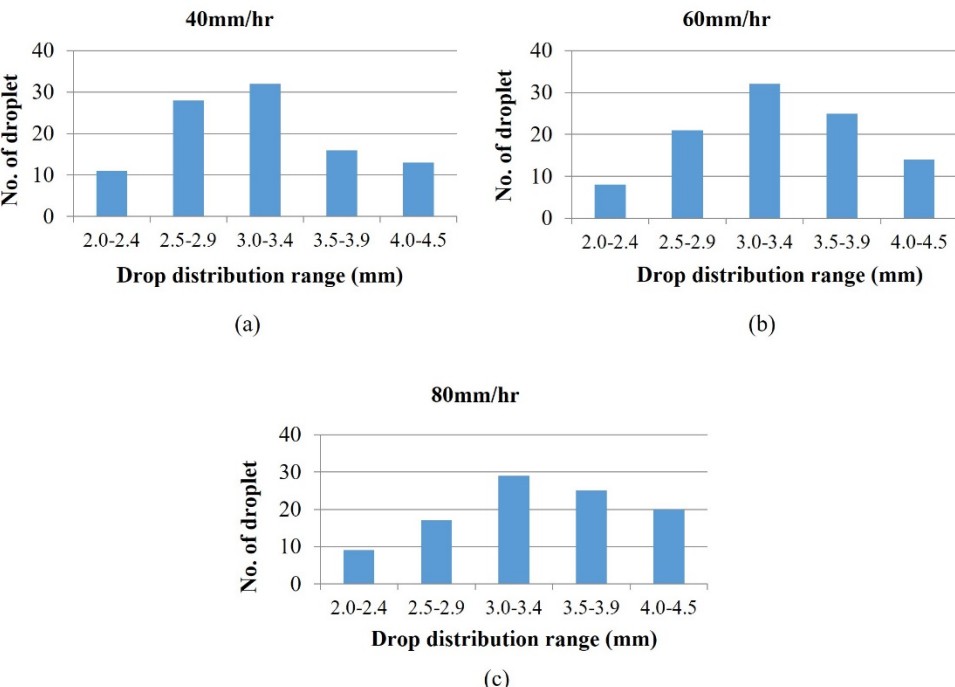

**Figure 4: Drop size distribution of the rain droplets for different rainfall intensities (a) 40mm/hr (b) 60mm/hr (c) 80mm/hr**


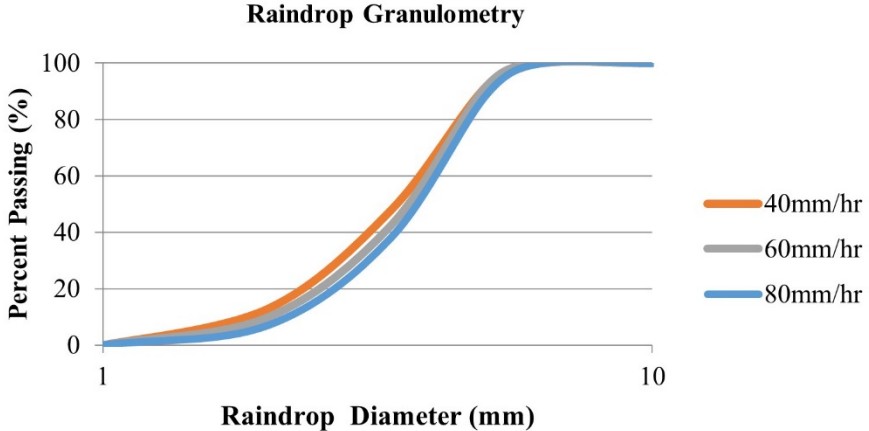

**Figure 5: Raindrop granulometry of different intensity**

Drop Size Distribution (DSD) is commonly expressed using percentages of values higher than the cumulative mass. $D_{50}$ is the
median value of the drop size distribution, and it is defined as half of the diameter of raindrops higher than the median value
of drop diameter. A rainfall event having high $D_{50}$ values shows an extreme rainfall event. During a rainfall event, drops are
measured, categorised and analysed according to the size, and $D_{50}$ are determined (Figure 6). According to Suhaila & Jemain
(2012), high-intensity events accounted for less than 11% of total rainy days in Peninsular Malaysia. However, this small
percentage contributed up to 50% of total rainfall intensities. $D_{50}$ values higher than 3.0 mm corresponds to rainfall intensities
higher than 60 mm/hr (Suhaila & Jemain, 2012). The values of $D_{50}$ in this study are 3.4, 3.6, and 3.7 for 40, 60 and 80 mm/hr,
respectively. Therefore, the produced rain falls in relatively large-sized drops. Substantial raindrop sizes contribute to high
kinetic energy, and erosivity can lead to soil erosion. From Figure 6, a positive correlation between rainfall intensity and $D_{50}$,
whereby an increase in rainfall intensity leads to an increase in $D_{50}$.

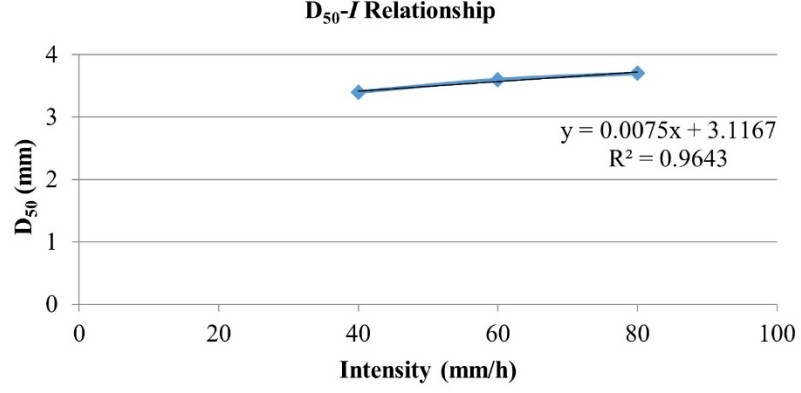

**Figure 6: The relationship of $D_{50}$ and rainfall intensity**



- **Uniformity of Simulated Rainfall**

Drip simulator was designed to ensure that the water is distributed efficiently and uniformly. The uniformity describes how evenly an irrigation system distributes water. It is regarded as an important feature for selecting, designing, and managing the irrigation system. The manufacturers' coefficient of variation ($Cv$) was determined to decide whether the system was excellent, good and marginal. Results show (Table 2) that $Cv$ of 0.07 and emitter flow variation ($Q_{var}$) of 9.09% for 30 mm/hr emitter was found maximum at 5.1 m operating pressure and minimum $Cv$ of 0.02 for 25.5 m operating pressure. Meanwhile, for 25.5 m operating pressure, it was found that $Q_{var}$ is 8.05%. The coefficient of variation decreased as operating pressure increased, and based on the results ($Cv$), all drip at different operating pressure falls under 0.02 to 0.07. Hence it can be concluded that drippers are in a good type where the excellent performance should be less than 0.05.

**Table 2: Uniformity parameter of emitters at different operating pressures.**

| Pressure (m) | Rainfall Intensity mm/hr | $qa$ (m³/s) | $Sq$ (m³/s) | $CU$ (%) | $Cv$ | $EU$ (%) | $DU$ (%) | $Q_{var}$ (%) |
|---|---|---|---|---|---|---|---|---|
| 5.1 | 30 | $8.65 \times 10^{-8}$ | $6.05 \times 10^{-8}$ | 97.58 | 0.07 | 96.42 | 96.38 | 9.09 |
| 10.2 | 40 | $1.16 \times 10^{-7}$ | $4.91 \times 10^{-9}$ | 96.74 | 0.05 | 95.92 | 94.72 | 13.64 |
| 15.3 | 60 | $1.71 \times 10^{-7}$ | $2.94 \times 10^{-9}$ | 98.54 | 0.04 | 97.24 | 98.33 | 4.76 |
| 20.4 | 70 | $2.09 \times 10^{-7}$ | $6.70 \times 10^{-9}$ | 97.51 | 0.03 | 94.16 | 96.37 | 10.13 |
| 25.5 | 80 | $2.33 \times 10^{-7}$ | $5.31 \times 10^{-9}$ | 98.40 | 0.02 | 95.35 | 97.34 | 8.05 |

*$qa$, average emitter discharge rate ; $Sq$, standard deviation of emitter flow rate ; $CU$, Christiansen's uniformity coefficient ; $Cv$, Variation coefficient of emitter flow rate; $EU$, Emission uniformity; $DU$, Low quarter distribution uniformity, $Q_{var}$, emitter flow variation.

Emission uniformity ($EU$) is one of the most frequently used in design criteria for the drip simulator systems where it is used to describe the emitter flow variation along a lateral line (Sarker et., al, 2019). Emission uniformity measures the uniformity of emitters discharge from all the emitters of drip irrigation system and is the most important parameter for evaluating system performance. Emission uniformity, $EU$ of the system decides the uniformity distribution of discharge by each emitter or uniformity distribution of water to each crop. The $EU$ shows a relationship between the minimum and average emitter discharge. The calculated emission uniformity data at different pressure (Table 3) was in a range of 95 to 97%. The general criteria for $EU$ are 90% or greater is considered as an excellent performance. The calculated coefficient of uniformity shows 96% to 99% results, which shows that the results are categorised as the water distributed uniformly. The general criteria for an excellent drip performance are more than 90%.



**Table 3: Terminal Velocity of raindrops**

| Rainfall Intensity (mm/hr) | Terminal Velocity (m/s) |
|---|---|
| 40 | 8.45 |
| 60 | 8.59 |
| 80 | 8.65 |

• **Drop velocity**

Equation 3 is used to calculate the terminal velocity to confirm that raindrops reach their terminal velocity immediately when they reach the flume soil. The velocity for the median drop size of 40, 60 and 80 mm/hr are shown in Table 3. The increase in raindrop diameter influenced the terminal velocity. The gap between the drop and the soil surface plot in the rain simulator set-up is set at 1.5 m. The result shows that various droplets velocities show that different drop sizes reached their terminal velocities after travelling various heights. It is also clear that, depending on the size of the droplets, the velocity with which

they strike the soil surface is less or more than their corresponding terminal velocities.. For a droplet to strike the soil surface with its terminal velocity the height of the experimental set-up needs to be greater, which is not feasible for laboratory experiments.

According to the results, for 40 mm/hr intensity, drop size in the range of 2.0-2.4 mm that constitute 11% of the total drop,

reached 75% of their terminal velocity. For drops in the size range of 2.5- 2.9 mm, 28% of the generated drops reached 70% of their terminal velocity and 3.0- 4.5 mm drops, which constitute 61% of the total drop, reached 63% of their terminal velocity. For 60 mm/hr intensity, it was found that those in the range 2.0-2.4 mm diameter, which constitutes 8% of the total drop, reach 75% of their terminal velocity. For drop with the size range of 2.5- 2.9 mm, which were 21% of the total drop, reach 70% of their terminal velocity; and 3.0- 4.5 mm drops, which constitute 71% of the total drop, reach 63% of their terminal velocity.

Lastly, for 80 mm/hr intensity, drops in the size 2.0-2.4 mm reach 75% of their terminal velocity, which constitute 9% of the total drop; then those in the range 2.5-2.9 mm, which constitute 17% of the generated drops, reach 70% of their terminal velocity. The diameter drop range 3.0- 4.5 mm, which constitute 74% of the total drop, reach 63% of their terminal velocity.

It is expected in rainfall simulations at lower height, raindrops plummet to the ground at lower terminal velocity. Some cases

include drip-type field simulators and most laboratory-type simulators unless it is located in stairwells or specifically-built towers (e.g. Cerdà et al., 1997;Clarke & Walsh, 2007). Epema and Riezebos (1983) presented their result with the mean velocity at the ground for 4.1 mm diameter raindrops falling from 0.85 m is 4.7 m/s, as compared with a terminal velocity by Laws (1941) for the same size of the drop of 9m/s (Clarke & Walsh, 2007). For simulation where the fall height is limited, such as on-site portable simulators or small-scale laboratory simulators, a larger median drop size is used to compensate for

the velocity reduction. According to Clarke & Walsh (2007), at 1.5 mm, with a terminal velocity of 5.5 m/s, Hudson (1963) obtained the median drop diameter of natural rainfall at intensities greater than 165 mm/hr (Laws, 1941). Therefore, from this study, the 3.00 – 3.40 mm values of $D_{50}$ produced by the simulator creates acceptable high drop velocity despite the low fall height.

• **Kinetic energy**

Kinetic Energy was estimated using the terminal velocity of raindrops using the Uplinger (1981) exponential formula, a close approximation of Gunn & Kinzer (1949) terminal velocities. The Kinetic Energy ($KE$) values per unit area per unit rain depth (J/m$^2$mm) were calculated from Equation (5) and are presented in Figure 7.

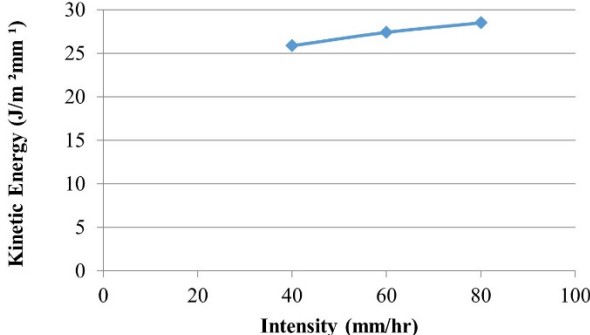

**Figure 7: Observation of kinetic energy vs intensity**

The $KE$ ranges from as low as 25.88 J/m$^2$mm followed by 27.42 J/m$^2$mm to 28.51 J/m$^2$mm, corresponding to 40, 60 and 80 mm/hr rain intensities. The high value of $KE$ emulates that rainfall with high intensity exceeds the impacts of lower raindrop fall heights and velocities. Lower $KE$ gives the impression of a shorter storm duration with lower fall heights. This exhibits 255 how high rainfall intensities contribute to more realistic $KE$ rates and the total $KE$ to be achieved. Van Dijk et al (2002) presented that $KE$ ranged from 11.00 J/m$^2$mm to 36.00 J/m$^2$mm for measured rainfall storms worldwide. This study replicated the rainfall $KE$ through the simulator as the $KE$ values presented are within the reported range.

**5 Conclusion**

A rainfall simulator was designed, calibrated, and tested to produce heavy rainfall intensities (40, 60 and 80 mm/hr), meeting 260 all of the major requirements formulated at the beginning of the conceptual planning. The drop size distributions were obtained by using a flour pellet methodology, described by Kathiravelu et.al (2016). The experiments were carried out with 100 drips under a series of operational pressures. Natural rainfalls are hard to replicate, but based on this study, with a systematic design of drip emitter and precise control of variable application rates, a rain simulation that produces rainfall attributes similar to natural rainfall is achieved. The results discussed in the previous section indicate that the fully automated rainfall simulation

system with precise control of investigates rates ranging from 40, 60 and 80 mm/hr. The drop size distribution is measured using the flour pellet method, where the produce rain shows an average drop size of 3.0-3.4 mm for all investigate rainfall intensities. The rainfall simulation system was installed 1.5 m in height, allowing 3.0 mm rain droplets and smaller that reach 63-75% of their respective terminal velocities. The kinetic energy produced by the rain simulator ranges from 25.88 - 28.51 J/m²mm, which is consistent with reported values worldwide. The results from this study show that Kinetic energy and

Terminal velocity was dependent on DSD. Finally, it can be concluded that the rainfall characteristic meets most of the desired standards concerning homogeneity and intensity. This study provides a positive error culture when discussing rainfall simulator specifications to acknowledge inevitable errors that might occur. However, these errors have to be minimised as far as possible.

**Author Contribution**

Harris Ramli conceptualized and edited the paper, Siti Aimi Nadia Mohd Yusoff conducted the tests and analysis of the paper,
Mastura Azmi wrote the final draft of the paper. Nuridah Sabtu verified and check the analysis and Muhd Azril Hezmi reviewed the draft.

**Competing Interest**

The authors declare that they have no conflict of interest

**Acknowledgement**

Authors would like to acknowledge the Ministry of Higher Education Malaysia for Fundamental Research Grant Scheme with Project Code: FRGS/1/2020/TK0/USM/02/19

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
