# Peer review of "Influence of drop size distribution and kinetic energy in precipitation modelling for laboratory rainfall simulators"

_Hydrology and Earth System Sciences, 2021_

## Referee Comment (RC2)

[referee-annotated manuscript omitted]

---

## Author Comment (AC1)

The authors wish to thank the editorial team for managing the technical review. The authors also thank the reviewers for the comments and appreciate their support in making the paper more readable and more beneficial. Notes: Authors' responses are highlighted in red in the revised manuscript.

**Answer to the comments of Reviewer 1**

| No. | Comment | Action | Notes |
|---|---|---|---|
| 1 | The authors are kindly advised to revise the article as the article as presented does not allow, in its current form the readers of HESS to use the results / the rainfall simulator discussed in this work. Major revision and/or rewriting of the article is needed to clear up parts that are unclear and to add information that is currently missing. | They are considered in the revised manuscript. | The authors thank you for the comments and appreciate the support and involvement of the reviewers to improve the paper. |
| 2 | A photo of the system needs to be added. The text mentions that every lateral contains 10 drip emitter at 0.8 meter, which would make the system 8 meters long. I'm assuming that this is 0.08m, as per the drawing in figure 1b. | Regarding this comment, the author changed "0.8 m" to "8 cm", similar to the drawing in Figure 1b. | Thanks to the reviewer for this comment. |
| 3 | The system seems to be 80 x 80 cm of nozzles. The text says that the flour pallet method is used to determine drop sizes. | Regarding this comment, the author rewrites the sentence to provide a more clear view, where the sentences have been modified to read as follows:

The system consists of 100 nozzle drip points distributed throughout a 72 cm × 72 cm square area. Ten lateral lines connected to the mainline contain ten nozzle drips equally spaced at 8 cm. The system's effective test area is 80 cm × 80 cm. | The sentences have been revised to avoid confusion with Figure 1b, as the lateral flow line spans over one metre (72cm + 39cm), with the drip spaced at an 8cm interval. Section 3.2 Drop Sizes elaborated on the method of using a flour pallet. |
| 4 | The author make no mention of the 'edge effects'? Did all the rain fall within the 1 square meter of the board? Of so: was it uniform? | Regarding this comment, the author added a paragraph on rainfall simulator calibration to provide a more clear view:

The rainfall simulator was calibrated in terms of rainfall intensity to achieve reproducible and consistent rainfall characteristics. Throughout the calibration | Thanks to the reviewer for this comment and suggestion. |

| | | and experiment, any element that may affect the changes in wind flow, such as an air conditioner or fan, is controlled to ensure that the raindrop falls vertically within the effective test area. The calibrating process was split into two parts. The first step measured the rainfall intensity and spatial rainfall distribution on the designated surface. To obtain high-resolution datasets, 100 cylindrical rainfall collectors are positioned on an area of 80 cm x 80 cm under the drip used. Each collector was weighed to determine the amount of rain collected after one hour of simulated rainfall. Based on the observations, no droplet crosses the cylindrical during the calibration and all land directly on the cylindrical beneath the drip. This initial calibration step ensures that each nozzle produces an equal amount of rainfall. A second step involved using a single large, plot-sized collection to determine net rainfall intensities. The volumetric method of flow measurement was utilised to calibrate the simulated real intensity. Two laboratory steel trays with top dimensions of 80cm x 80cm x 10cm and bottom dimensions of 100cm x 100cm x 8cm were used for volume control and placed beneath drip systems. The different heights of the top and bottom are merely a coincidence due to the laboratory's available tray. The primary technical requirement is that the central collection tray (small) must fit within the dimensions of the designed test area and be tall enough to collect the intended rainfall intensity (in this work, the maximum is 80 mm/h.), whereas the secondary collection tray must be larger than the effective test area. Collector boxes were placed in a central location (in relation to the drip location) and collected the precipitated volume at a set pressure. The volume of precipitated water was determined using a measuring cylinder. A ruler was used to measure the water level, and then the precipitation volume collected was recorded. No raindrops landed on the large bottom tray during the second calibration stage based on the observations. This demonstrates that the raindrop area's uniformity is reproducible. | |
| 5 | How was the calculation from water flow to mm/hour done? | Regarding this comment, the author added sentences to provide a more clear view:

The calibration result enables the flowmeter to correlate the amount of water controlled | Thanks to the reviewer for this comment and suggestion. |

| | | | |
|---|---|---|---|
| | | by the flowmeter to the amount of water emitted from the nozzle. Thus, the simulator delivers the desired rainfall intensity (mm/hr). | |
| 6 | Figure 1a suggest that the 'soil' is uneven and thus that different heights are possible in this system. If this is true it needs to be made explicit. If it is not, figure 1a needs to be corrected | No changes in manuscript | Thanks to the reviewer for this comment. The uneven soil profiles result from another aspect of this research that is not covered in this paper. Calibration results indicated that small changes to these soil profiles do not affect the rainfall simulator's performance. |
| 7 | The authors do not mention in their introduction what range their new simulator targets, nor for which applications it is build. | No changes in manuscript | Thanks to the reviewer for this comment and suggestion. The main intention of this simulator setup is to simulate targeted rainfall intensities similar to the cases of landslide occurrences in the authors country. The simulator is applied to a laboratory setup of a specific selection of slope criteria for further understanding on rainfall impact in groundwater level changes. It is beyond this manuscript context; therefore, the authors excluded the explanation of the simulator's application. |
| 8 | Paragraph 3 contains a mix of theory and experimental setup that is hard to disentangle. The choices made in how to conduct the experiment are mentioned in between citations to literature. I strongly recommend separating paragraph 3 in a 'theory' and a 'experimental setup' paragraph. | No changes in manuscript | Thanks to the reviewer for this comment and suggestion. The third paragraph discussed solely with the subject's literature. |
| 9 | To check the amount of water coming out of the system the flow meter is read for different settings of the pump pressure. It is not mentioned how this experiment was | As mentioned in the previous comment (No. 4 and No. 5), the author added a paragraph on rainfall simulator calibration to provide a clearer view. | Thanks to the reviewer for this comment and suggestion. |

| | | | |
|---|---|---|---|
| | conducted: for how long was the valve opened? Was the flow allowed to settled before starting the measurement? Or was it the same 1 to 4 seconds mentioned in paragraph 3.2? | | |
| 10 | To check the drop sizes generated with this system, the flour pallet method was used with the flow opened for 1 to 4 seconds (how much? How was this determined? Was the idea to only have a few drops? How many?) This needs clearing up on what was done and how it relates to what is reported. | Regarding this comment, the author added sentences to provide a more clear view:

According to Kathiravelu et al., 2016, the flour pallet test would be conducted by various researchers between 1 and 4 seconds, depending on the intensity of rainfall. In our experiment, a 1.0 m x 1.0 m plate containing a 2.54 cm (1 inch) layer of uncompacted fine wheat flour was exposed to a split second of rainfall. The drops must not fall at the same point during rainfall simulation and formed corresponding to the drop size on impact. The objective is to collect a single drop of water. The flour plate was positioned 1.5 metres below the drip and covered with two layers of tray. A collection tray is used on both the top and bottom trays. The top collection tray's purpose is to capture the initial 1 to 2 seconds of rainfall when the valve is opened. This ensures that every nozzle produces a raindrop. Simultaneously with the top tray being set aside, the flour plate was exposed for a split second before being covered by the bottom tray. The valve was immediately closed to halt the rain. Based on the observation, the raindrop pattern is within the effective test area. | Thanks to the reviewer for this comment and suggestion. |
| 11 | The authors indicate they open and close the valve for 1 to 4 seconds. Is this how it will always be operated? Or will it be opened fully when used in practice? | No changes in manuscript | Thanks to the reviewer for this comment and suggestion. As explained in comment No. 10, the statement of 1 to 4 seconds is for the flour pallet test only. During the actual rainfall simulation test, the intended volume of rainfall is applied according to the rainfall intensity and duration. |

---

## Author Comment (AC2)

The authors wish to thank the editorial team for managing the technical review. The authors also thank the reviewers for the comments and appreciate their support in making the paper more readable and more beneficial. Notes: Authors' responses are highlighted in red in the revised manuscript.

**Answer to the comments of Reviewer 2**

| No. | Comment | Action | Notes |
|---|---|---|---|
| 1 | The authors are kindly advised to revise the article as the authors first fail to show any improvement of their proposed solution over those already available in the literature, starting from the ones presented in papers that are mentioned among the references of this manuscript. There is no evidence in the manuscript, indeed, that the presented setup performs any better than very similar solutions that already exists. | They are considered in the revised manuscript. | The authors thank you for the comments and appreciate the support and involvement of the reviewers to improve the paper. The novelty of this paper compared to Mendes (2021), is that this paper proves that the usage of drip emitter for a high rainfall intensity is more suitable than spray nozzle. This paper also proves that raindrops' size distribution is more consistent when using drip emitter. This can be seen from consistent median value in the normal distribution of $D_{50}$. This paper also proves that the collection area is important in rainfall simulator selection (drip emitter vs. spray nozzle). |
| 2 | The main comments reported in the conclusions do not reflect results of the present manuscript | Terminal velocity is calculated based on the equation (3) – Text in abstract needs to change from monitored to calculated

RC2: the main comments reported in the conclusions do not reflect results of the present manuscript (e.g., no calibration exercise is described in the text, no validation with real rainfall characteristics is provided, etc.). General statements about the dependence of the drop size distribution and the kinetic energy of the raindrops on the rainfall intensity are declared as results of the present study, while obviously they are not. | |
| 3 | This is correctly states as a requirement but not demonstrated in the manuscript.

"Surely, simulated rainfall should made to be as close to natural rainfall as possible, especially on droplets' fall velocity and size distribution." | No changes in manuscript | Thanks to the reviewer for this comment. The main novelty of our paper is that we prioritise improving the simulated rainfall droplet size distribution, as this is something that other researchers can effectively replicate in comparison to the fall velocity parameter. |

| 4 | Correction of the terminology "…intensity of the raindrop…" | Regarding this comment, the author had changed "raindrop" to "rainfall". | Thanks to the reviewer for this comment and suggestion. |
|---|---|---|---|
| 5 | This is not demonstrate in the manuscript.

"…capable of producing droplet characteristics identical to natural rainfall." | Regarding to this comment, the author adds more detailed information on this sentence to give a clearer view, where the sentences has been modified to:

"…capable of producing droplet characteristics identical to those found in natural rainfall from a drop size distribution perspective." | Thanks to the reviewer for this comment and suggestion. |
| 6 | Correction of the terminology.

"The pipe diameter is 2 cm while the spray diameter is 4 mm." | Regarding this comment, the author changed "spray" to "nozzle". | Thanks to the reviewer for this comment and suggestion. |
| 7 | It is also important to note that high rainfall intensities degrade the quality of measurements using the disdrometers or imaging techniques, due to background noise (Mendes et al., 2021). | No changes in manuscript | Thanks to the reviewer for this comment and suggestion. This sentence intends to emphasise that this study, like Mendes et al. 2021, is not suitable for using disdrometers due to the high rainfall intensity (40 mm/hr to 80 mm/hr). |
| 8 | Was this calibrated in some way or is it a literature result? $D_d = 14.56 M_f^{0.354}$ and $D_d$ (mm) $= 0.985 D_f$ (mm)$^{1.02}$ | Regarding this comment, the author omitted the sentences:

The relationship between the diameter of the drop, $D_d$ (in mm), and the mass of the flour ball, $M_f$ (in mg), is $D_d = 14.56 M_f^{0.354}$, respectively. To estimate the drop diameter as a function of the flour ball diameter $D_d$ (mm) $= 0.985 D_f$ (mm)$^{1.02}$, the diameter of flour balls can be considered practically equivalent to that of the drops. | Thanks to the reviewer for this comment and suggestion. The sentences are omitted because the formula is a literature result based on Navas et al., 1990. In our case, we might not need to use the formula since the drop diameter is measured, and the result shows a consistent median value in the normal distribution of $D_{50}$. |
| 9 | No measurements of fall velocity and kinetic energy are reported | Regarding this comment, the author change the typo "$I$ = Rainfall Intensity," to the "$I$ = Rainfall Intensity." and missing words added to the sentences:

$I$ = Rainfall Intensity. Kinetic energy can also be measured from the physical properties of simulated rainfall: raindrop size, fall velocity and drop size distribution. | Thanks to the reviewer for this comment and suggestion. |

| 10 | Why is this different from Eq. 3 used above ? | Regarding this comment, the Equation is same as the Eq.3 | Thanks to the reviewer for this comment and suggestion. |
|---|---|---|---|
| 11 | Could not see any calibration exercise described in this paper. What is the meaning of "calibrated" here ? | Regarding this comment, the author added a paragraph on rainfall simulator calibration to provide a more clear view:

The rainfall simulator was calibrated in terms of rainfall intensity to achieve reproducible and consistent rainfall characteristics. Throughout the calibration and experiment, any element that may affect the changes in wind flow, such as an air conditioner or fan, is controlled to ensure that the raindrop falls vertically within the effective test area. The calibrating process was split into two parts. The first step measured the rainfall intensity and spatial rainfall distribution on the designated surface. To obtain high-resolution datasets, 100 cylindrical rainfall collectors are positioned on an area of 80 cm x 80 cm under the drip used. Each collector was weighed to determine the amount of rain collected after one hour of simulated rainfall. Based on the observations, no droplet crosses the cylindrical during the calibration and all land directly on the cylindrical beneath the drip. This initial calibration step ensures that each nozzle produces an equal amount of rainfall. A second step involved using a single large, plot-sized collection to determine net rainfall intensities. The volumetric method of flow measurement was utilised to calibrate the simulated real intensity. Two laboratory steel trays with top dimensions of 80cm x 80cm x 10cm and bottom dimensions of 100cm x 100cm x 8cm were used for volume control and placed beneath drip systems. The different heights of the top and bottom are merely a coincidence due to the laboratory's available tray. The primary technical requirement is that the central collection tray (small) must fit within the dimensions of the designed test area and be tall enough to collect the intended rainfall intensity (in this work, the maximum is 80 mm/h.), whereas the secondary collection tray must be larger than the effective test area. Collector boxes were placed in a central location (in relation to the drip location) and collected the precipitated volume at a set pressure. The volume of precipitated water was determined using a measuring cylinder. A ruler was used to | Thanks to the reviewer for this comment and suggestion. |

| | | | |
|---|---|---|---|
| | | measure the water level, and then the precipitation volume collected was recorded. No raindrops landed on the large bottom tray during the second calibration stage based on the observations. This demonstrates that the raindrop area's uniformity is reproducible. | |
| 12 | Repeated sentence | Regarding this comment, the author omitted a sentence to avoid repetition in conclusion:

The drop size distributions were obtained by using a flour pellet methodology, described by Kathiravelu et.al (2016). | Thanks to the reviewer for this comment and suggestion. |
| 13 | This was not demonstrated in the paper. | Regarding this comment, the author added a sentence to provide a more specific conclusion:

…natural rainfall is achieved from the perspective of rainfall intensity and drop size distribution. | Thanks to the reviewer for this comment and suggestion. |
| 14 | The results presented in the manuscript do not show that. It was simply assumed from literature results. | Regarding this comment, the author rewrite a sentence to provide a more specific conclusion:

This study discovered that DSD might affect both kinetic energy and terminal velocity. | The sentences have been rewritten so that they are less ambiguous. |